# Abiotic formation of alkylsulfonic acids in interstellar analog ices and implications for their detection on Ryugu

Mason McAnally [1,2,4], Jana Bocková [3,4], Ashanie Herath [1,2], Andrew M. Turner[1,2], Cornelia Meinert [3] ✉ & Ralf I. Kaiser [1,2] ✉

For the last century, the source of sulfur in Earth's very first organisms has remained a fundamental, unsolved enigma. While sulfates and their organic derivatives with sulfur in the S(+VI) oxidation state represent core nutrients in contemporary biochemistry, the limited bioavailability of sulfates during Earth's early Archean period proposed that more soluble S(+IV) compounds served as the initial source of sulfur for the first terrestrial microorganisms. Here, we reveal via laboratory simulation experiments that the three simplest alkylsulfonic acids—water soluble organic S(+IV) compounds—can be efficiently produced in interstellar, sulfur-doped ices through interaction with galactic cosmic rays. This discovery opens a previously elusive path into the synthesis of vital astrobiological significance and untangles fundamental mechanisms of a facile preparation of sulfur-containing, biorelevant organics in extraterrestrial ices; these molecules can be eventually incorporated into comets and asteroids before their delivery and detection on Earth such as in the Murchison, Tagish Lake, and Allende meteorites along with the carbonaceous asteroid Ryugu.

While sulfates ($SO_4^{2-}$) and their organic derivatives represent core (in)organic nutrients utilized in contemporary biochemistry[1], the prevalence of only low concentrations of sulfates during Earth's early Archean period suggests a limited bioavailability of vital sulfates[2,3]. Sulfur in the +IV oxidation state presents an alternate source of sulfur for the first terrestrial microorganisms[4]. Alkylsulfonic acids (ASAs, $RSO_2(OH)$ where R represents an alkyl group, Fig. 1) typifies the S(+IV) oxidation state of sulfur present in key biological systems such as coenzyme M ($HSCH_2CH_2SO_2(OH)$), which is utilized by methanobacteria[5], and taurine ($H_2NCH_2CH_2SO_2(OH)$), i.e., a prominent molecular component in bile[6] and energy source of prokaryotes[7]. Meteoritic sulfides such as pentlandite $(Fe, Ni)_9S_8$ and troilite (FeS) identified in the Murchison meteorite[8–10] have been studied extensively as a potential source of sulfates ($SO_4^{2-}$) and sulfites ($SO_3^{2-})$[11–13]. While the alteration of meteoritic sulfides centers on the terrestrial

transformation of sulfur to produce biologically relevant material, alternatively, sulfur-bearing molecules such as ASAs could have been formed in astrophysical environments prior to their delivery to Earth[14].

The identification of a reduced form of S(+IV) as highly water-soluble ASAs, including methylsulfonic acid (MSA, $CH_3SO_2(OH)$), ethylsulfonic acid (ESA, $CH_3CH_2SO_2(OH)$), n/i-propylsulfonic acid (n/i-PSA, $C_3H_7SO_2(OH)$), as well as larger ASAs up to C14 carbon-atoms[15] in meteorites such as Murchison, Tagish Lake, and Allende, and more recently in samples from the carbonaceous asteroid Ryugu[16], affords an extraterrestrial basis for prebiotic sulfur chemistry. These ASAs could supply a vital source of organosulfur molecules during the earliest stages of biochemical evolution on Earth and could have acted as precursors to key classes of molecules such as sulfoquinovose ($C_6H_{12}O_8S$) and 3'-phosphoadenosine-5'-phosphosulfate ($C_{10}H_{15}N_5$ $O_{13}P_2S$), which retain biochemical functions as sugars and coenzymes,

[1]Department of Chemistry, University of Hawaii at Mānoa, Honolulu, HI, USA. [2]W.M. Keck Laboratory in Astrochemistry, University of Hawaii at Mānoa, Honolulu, HI, USA. [3]Université Côte d'Azur, Institut de Chimie de Nice, UMR 7272 CNRS, Nice, France. [4]These authors contributed equally: Mason McAnally, Jana Bocková. ✉e-mail: Cornelia.MEINERT@univ-cotedazur.fr; ralfk@hawaii.edu

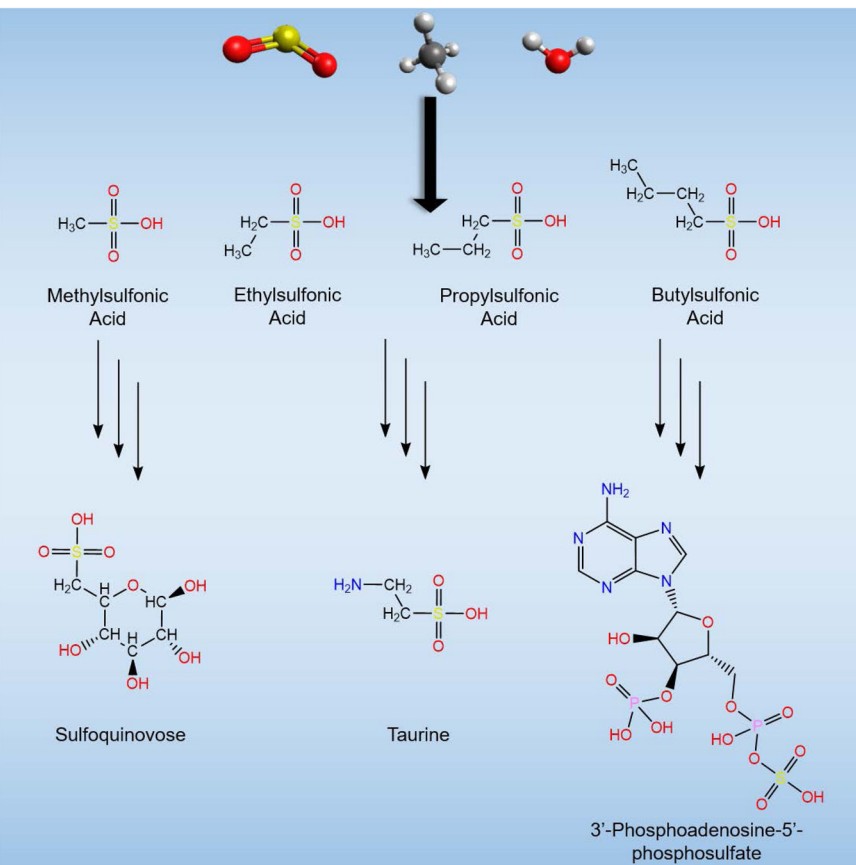

**Fig. 1 | The role of alkylsulfonic acids in contemporary biochemistry.** Sulfur (yellow), oxygen (red), carbon (gray), and hydrogen (white) form the core components of alkylsulfonic acids (RSO$_2$(OH) where R is an alkyl group) which are critical to contemporary biochemistry and represented in biology as sulfur-containing sugars, amino sulfonic acids, and sulfur-oxygen bonding found in sulfotransferase coenzymes.

respectively. Laboratory simulation experiments demonstrated that biologically relevant molecules like amino acids and sugars can be synthesized abiotically via processing of low temperature interstellar ice analogs on interstellar nanoparticles (interstellar grains) at temperatures as low as 10 K in cold molecular clouds by ionizing radiation (photons, energetic electrons)[17–19]. However, although sulfates (SO$_4^{2-}$) and sulfites (SO$_3^{2-}$) have been identified in these model ices[20–22], abiotic formation pathways leading to alkylsulfonic acids in low temperature extraterrestrial environments are still elusive. Since complex, bio-relevant organics found in chondrites likely originate from interstellar ices[17,19], it is plausible that also sulfur-bearing molecules were incorporated into minor celestial bodies, eventually being delivered to early Earth[14]. Sulfur dioxide (SO$_2$) presents a promising source of sulfur in interstellar grains and comets considering recent detections in the coma of 67 P/Churyumov-Gerasimenko[23] and by the James Webb Space Telescope[24] at levels of 0.05% compared to water (H$_2$O). Although SO$_2$ is a prominent source of sulfur in the ISM, it only contributes to a few percent of the total sulfur budget[25]. The remaining sulfur is characterized in several different forms including iron(II) sulfide (FeS), magnesium sulfide (MgS), carbonyl sulfide (OCS), and hydrogen sulfide (H$_2$S)[26]. Additionally, previous experiments have probed SO$_2$-containing ices[27–30] as well as sulfur atom implantation[31–33] to understand the underlying chemistry of sulfur in astrophysically relevant environments. When combined with methane (CH$_4$)—the simplest hydrocarbon in interstellar ices at concentrations up to 5% with respect to water[24,25]—the processing of sulfur dioxide by galactic cosmic rays (GCRs) at temperatures prevailing in cold molecular clouds (10 K) represent a viable starting point for the synthesis of ASAs in interstellar ices and their analogs.

We present here, surface-science experiments on the formation of the three simplest alkylsulfonic acids (MSA, ESA, *n*-PSA) in low-temperature interstellar analog ices carrying sulfur dioxide (SO$_2$), water (H$_2$O), and methane (CH$_4$). These experiments were carried out in an ultrahigh-vacuum chamber at pressures of a few $10^{-11}$ Torr by exposing ices to proxies of GCRs in the form of energetic electrons[34] at astrophysically relevant temperatures of 10 K. These conditions replicate the processing of interstellar ices in cold molecular clouds by galactic cosmic ray-initiated electron cascades over the lifetime of cold molecular clouds up to $5 \times 10^7$ years[35]. Galactic cosmic ray MeV particles penetrate through icy mantles and the grain core, depositing portions of their energy into interstellar grains. Simulations suggest the energy predominantly results in ionization, releasing energetic electrons born of a few keV[36,37]. These secondary electrons can penetrate the ice, initiating non-equilibrium chemical reactions. Although these electrons may not completely penetrate through the ice, they can process significantly more material. Here, the electron beam replicates these electron cascades found in various ages of molecular clouds: a low dose simulating $10^6$ years in cold molecular clouds (up to 4.2 eV molecule$^{-1}$), a medium dose ($10^7$ years; up to 42 eV molecule$^{-1}$), and a high dose ($5 \times 10^7$ years, up to 210 eV molecule$^{-1}$). Fourier transform infrared (FTIR) spectroscopy was utilized to monitor chemical changes in the ice during exposure to ionizing radiation and the temperature-programmed desorption (TPD) process in which the ice was heated from 10 K to 320 K. The TPD phase effectively simulated the increase of the temperature from a cold molecular cloud transitioning to star forming regions[35]. The solid residues remaining on the substrate surface onto which the interstellar analog ices have been condensed were analyzed after extraction and derivatization via two-dimensional gas

chromatography coupled with reflectron time-of-flight mass spectrometry (GC×GC–TOF-MS). The synthesis and detection of MSA, ESA, and n-PSA in interstellar analog ices offers compelling evidence that ASAs, such as those detected in Murchison, Tagish Lake, and Allende along with the carbonaceous asteroid Ryugu, can form in interstellar environments and could have been delivered to early Earth, thus acting as a plausible water-soluble source of organosulfur compounds hence promoting sulfur biochemistry in the early Solar System.

## Results

### Fourier transform infrared spectroscopy: SO$_2$/CH$_4$ system

Infrared spectroscopy of the irradiated ice allows for the identification of small molecules and functional groups of complex organic molecules[38]. The infrared spectra of the deposited sulfur dioxide/methane (SO$_2$:CH$_4$, 2:5) ices reveal prominent absorptions associated with the reagents such as the symmetric stretch ($\nu_1$, 1151 cm$^{-1}$), the bending mode ($\nu_2$, 524 cm$^{-1}$), and the asymmetric stretch ($\nu_3$, 1343 cm$^{-1}$) of sulfur dioxide along with the asymmetric stretching mode ($\nu_3$, 3012 cm$^{-1}$), symmetric stretch ($\nu_1$, 2906 cm$^{-1}$), and the deformation mode ($\nu_4$, 1306 cm$^{-1}$) for methane (Supplementary Fig. 1 and Supplementary Table 1)[21,29,39,40]. During the irradiation, prominent new features emerged (Fig. 2 and Supplementary Table 2). These can be linked to, e.g., carbon dioxide (CO$_2$) and carbon monoxide (CO) detected via the C=O asymmetric stretching mode ($\nu_3$, 2343 cm$^{-1}$) and bending mode ($\nu_2$, 668 cm$^{-1}$) for carbon dioxide and the carbon-oxygen stretching mode ($\nu_1$, 2138 cm$^{-1}$) for carbon monoxide[37]. Important intermediates to ASA, i.e. the methyl radical (CH$_3^{\cdot}$), could be traced via the CH stretching mode ($\nu_3$) at 3160 cm$^{-1}$, while the asymmetric stretching modes of sulfur trioxide (SO$_3$) were detectable at 1402 and 1382 cm$^{-1}$ [21,41]; broad and intense features in the 3500–3200 cm$^{-1}$ region reveals the formation of new hydroxyl groups.

Sharp absorptions in the CH stretching region (3000–2800 cm$^{-1}$) exhibit several new peaks and—coupled with the observation of the CH bending modes in the 1483–1434 cm$^{-1}$ range—support the formation of alkyl groups during the irradiation exposure[42]. Sulfuric acid could be identified through the hydroxyl (OH) bending modes at 1241 cm$^{-1}$, the S=O symmetric stretching at 1166 cm$^{-1}$, and SO stretching modes at 961 cm$^{-1}$ and 996 cm$^{-1}$ [43]. Most importantly, several peaks can be linked to functional groups associated with ASAs such as OH stretching modes at 2899 cm$^{-1}$ and 2523–2588 cm$^{-1}$, the CH bending mode at 1464 cm$^{-1}$, the S=O stretching mode at 1210 cm$^{-1}$, the SOH bending at 1119 cm$^{-1}$, CH rocking at 996 cm$^{-1}$, and the C–S stretching at 724 cm$^{-1}$ (medium dose) and 786 cm$^{-1}$ (high dose)[42–45]. Therefore, these data provide compelling evidence that functional groups related to ASAs originate during the radiation exposure at 10 K.

After the irradiation, the ices were warmed up to 320 K, and the resulting infrared spectra of the organic residues were probed spectroscopically (Fig. 3, Supplementary Table 3). These spectra exhibit functional groups associated with ASAs suggesting that at least a fraction of the functional groups linked to ASAs remains in the organic residues. Here, evidence for functional groups associated with ASAs is supported by the OH stretching at 2845 cm$^{-1}$ and 2426 cm$^{-1}$, the CH bending modes at 1416 (medium dose) and 1440 cm$^{-1}$ (high dose), the asymmetric S=O stretching at 1297 cm$^{-1}$, the SOH bending mode at 1186 cm$^{-1}$, the symmetric S=O stretching at 1116 cm$^{-1}$, SO stretching at 864 cm$^{-1}$, and the C-S stretching mode at 771 cm$^{-1}$ [42–45]. Experiments using isotopically-labeled reagents were conducted utilizing methane-$^{13}$C ($^{13}$CH$_4$) or methane-d$_4$ (CD$_4$). The SO$_2$/$^{13}$CH$_4$ infrared spectra (Supplementary Fig. 2 and Supplementary Tables 4-5) and SO$_2$/CD$_4$ infrared spectra (Supplementary Fig. 3 and Supplementary Tables 6-7) show similar products to the non-isotopically labeled experiment. Infrared band positions are redshifted when substituted

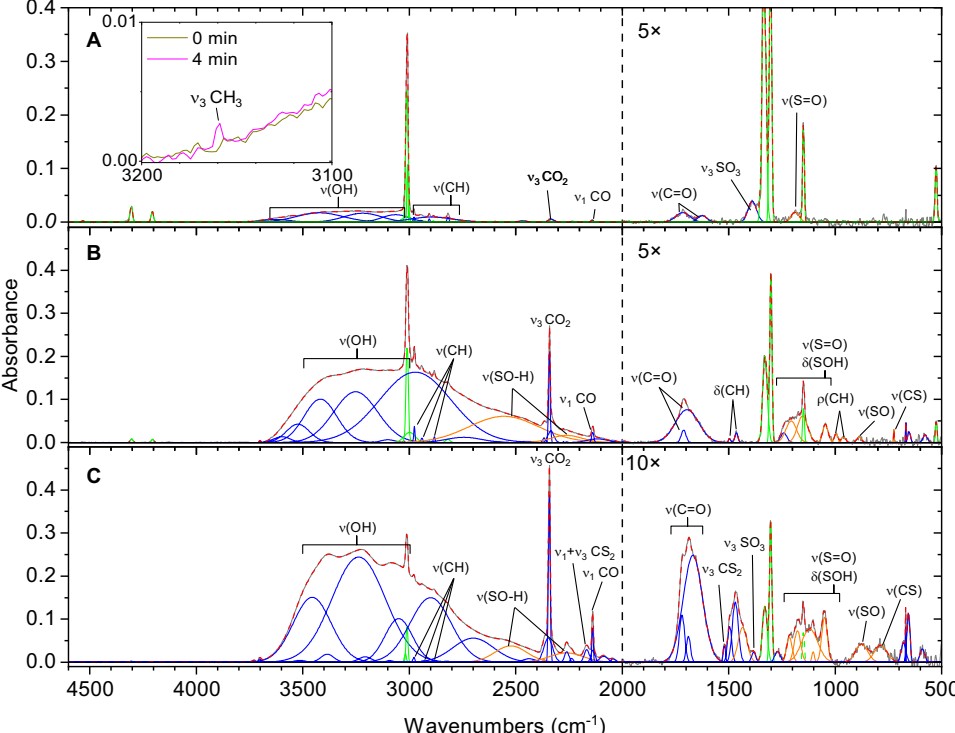

**Fig. 2 | FTIR spectra of sulfur dioxide (SO$_2$)/methane (CH$_4$) ices at 10 K after irradiation for 1 h.** Each spectrum was taken after an irradiation dose of **A** 100 nA, **B** 1000 nA, and **C** 5000 nA current. The original spectrum (gray) is deconvoluted showing functional groups associated with reagents (green), alkylsulfonic acids (orange), and irradiation products (blue). The red line represents the sum. The spectral region (2000–500 cm$^{-1}$) to the right of the dashed vertical line indicates the region has been magnified by 5 or 10 times for clarity. The inset depicts the ice sample prior to the irradiation (dark yellow) and after 4 min radiation exposure (purple).

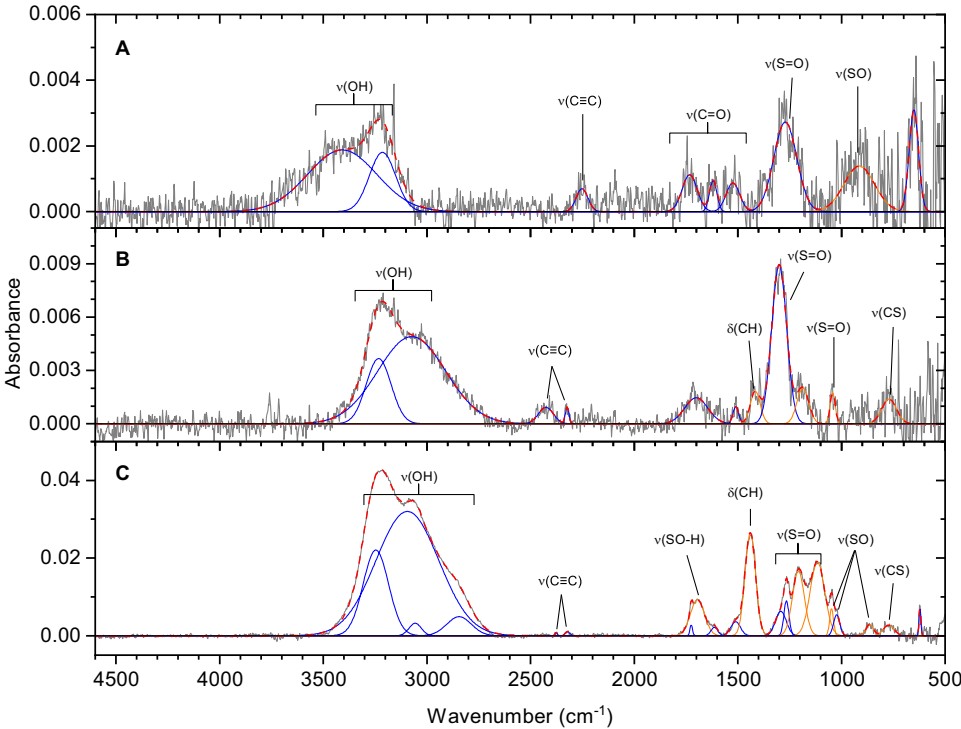

**Fig. 3 | FTIR spectra of sulfur dioxide ($SO_2$)/methane ($CH_4$) ice experiments at 320 K after irradiation for 1 hour.** These residues were probed after an irradiation dose of **A** 100 nA, **B** 1000 nA, and **C** 5000 nA current and warming to 320 K. The original spectrum (gray) is deconvoluted showing functional groups associated with alkylsulfonic acids (orange), and new irradiation products (blue). These deconvolutions sum to the peak fitted spectrum (red dashed).

with $^{13}$C. The peak at 1050 cm$^{-1}$ is observed in non-labeled and labeled experiments that suggest this functional group contains no carbon or hydrogen and, considering the region where the peak occurs, is confirmed to be SO stretching. Similarly, the lack of shifting of absorptions in the 1300–1100 cm$^{-1}$ region appears after irradiation and in the residue, confirming these peaks belong to the S=O functional group. Overall, our data afford persuasive testimony of the irradiation induced formation of functional groups associated with ASAs at 10 K.

**Fourier transform infrared spectroscopy: $SO_2$/$CH_4$/$H_2O$ system**
Since water represents the dominating component of interstellar ices, it is critical to also explore the interaction of ionizing radiation with water-rich ices containing sulfur dioxide and methane and to explore the effect(s) of water on the yields of ASAs[24]. Prior to the irradiation, the infrared spectrum depicts fundamentals of water in the 3600–3000 cm$^{-1}$ region ($v_1$/$v_3$), of the bending mode ($v_2$) at 1660 cm$^{-1}$, and the lattice vibration ($v_L$, 660–900 cm$^{-1}$) (Supplementary Fig. 4, Supplementary Table 8)[46]. Sulfur dioxide and methane exhibit identical features as in the anhydrous ices discussed above. During the irradiation, prominent products emerge (Supplementary Fig. 5 and Supplementary Table 9). These are carbon dioxide ($v_3$, 2342 cm$^{-1}$), carbon monoxide ($v_1$, 2137 cm$^{-1}$)[37], and ethane detected by $v_{10}$ at 2976 cm$^{-1}$, $v_5$ at 2882 cm$^{-1}$, and $v_8 + v_{11}$ at 2938 cm$^{-1}$ [39,47,48]. The formation of alkyl groups is supported through CH stretching vibrations increasing in the 3000–2800 cm$^{-1}$ region and the CH bending vibrations at 1486 and 1439 cm$^{-1}$. Most importantly, hydroxyl (OH) stretching modes related to ASAs are observed at 2996, 2497 (medium dose) and 2565 cm$^{-1}$ (high dose). ASAs are also supported via the CH bending mode at 1439 cm$^{-1}$, the S=O stretching mode at 1185 cm$^{-1}$, and the SOH bending mode at 1109 cm$^{-1}$ [42–45]. Finally, sulfuric acid can only be assigned by two peaks: the OH bending mode at 1270 cm$^{-1}$ and the S=O stretching at 1185 cm$^{-1}$ [143].

After the TPD to 320 K, the infrared spectra of the residues reveal consistent results from the low to the high dose studies

(Supplementary Fig. 6, Supplementary Table 10). The vibrational modes associated with the functional groups of ASAs are the CH bending mode at 1440 cm$^{-1}$, the S=O vibrations at 1320 and 1200 cm$^{-1}$, the SOH bending mode at 1125 cm$^{-1}$, the CH rocking at 963 cm$^{-1}$, and the SO stretch at 853 cm$^{-1}$ [42–45]. These results clearly reveal that functional groups of ASAs are also synthesized in the $SO_2$:$CH_4$:$H_2O$ (1:2:10) ices upon irradiation and warmup; however, the intensity of peaks likely associated with ASAs are an order of magnitude lower in the water-rich ice, which is consistent with lower quantities of limiting reagents—$SO_2$ and $CH_4$—required to produce ASAs. Additionally, radiolysis products derived from water may inhibit the formation of ASAs due to competing reactions consuming crucial intermediates. Since the infrared absorptions of individual ASAs fall in a similar range, and hence, overlap, infrared spectroscopy alone does not allow an identification of discrete and unique organic molecules such as ASAs. Therefore, additional analytical techniques are required to identify individual ASAs.

**Two-dimensional gas chromatography time-of-flight mass spectrometry**
The solid residues generated upon warming up of the irradiated samples to 320 K were analyzed for individual ASAs. The samples were first extracted with MilliQ® water, dried and derivatized prior to two-dimensional gas chromatography time-of-flight mass spectrometry (GC×GC–TOF-MS) analysis (Fig. 4, Supplementary Table 11). Each sample was compared against a procedural blank to confirm that the acids were formed during ice irradiation and to exclude contamination during storing or ex situ analysis.

Overall, the three simplest alkylsulfonic acids MSA, ESA and PSA, along with sulfuric acid ($H_2SO_4$) were detected through the comparison of dual retention times ($R_{t1} × R_{t2}$), the mass spectra of the *t*-butyldimethylsilyl derivative (M$^+$), and the subsequent *t*-butyl ($C_4H_9$) loss channel (M−57$^+$) with reference standards[17]. In detail, methylsulfonic acid (MSA) was identified via mass-to-charge (*m/z*) ratio 210 (M$^+$)

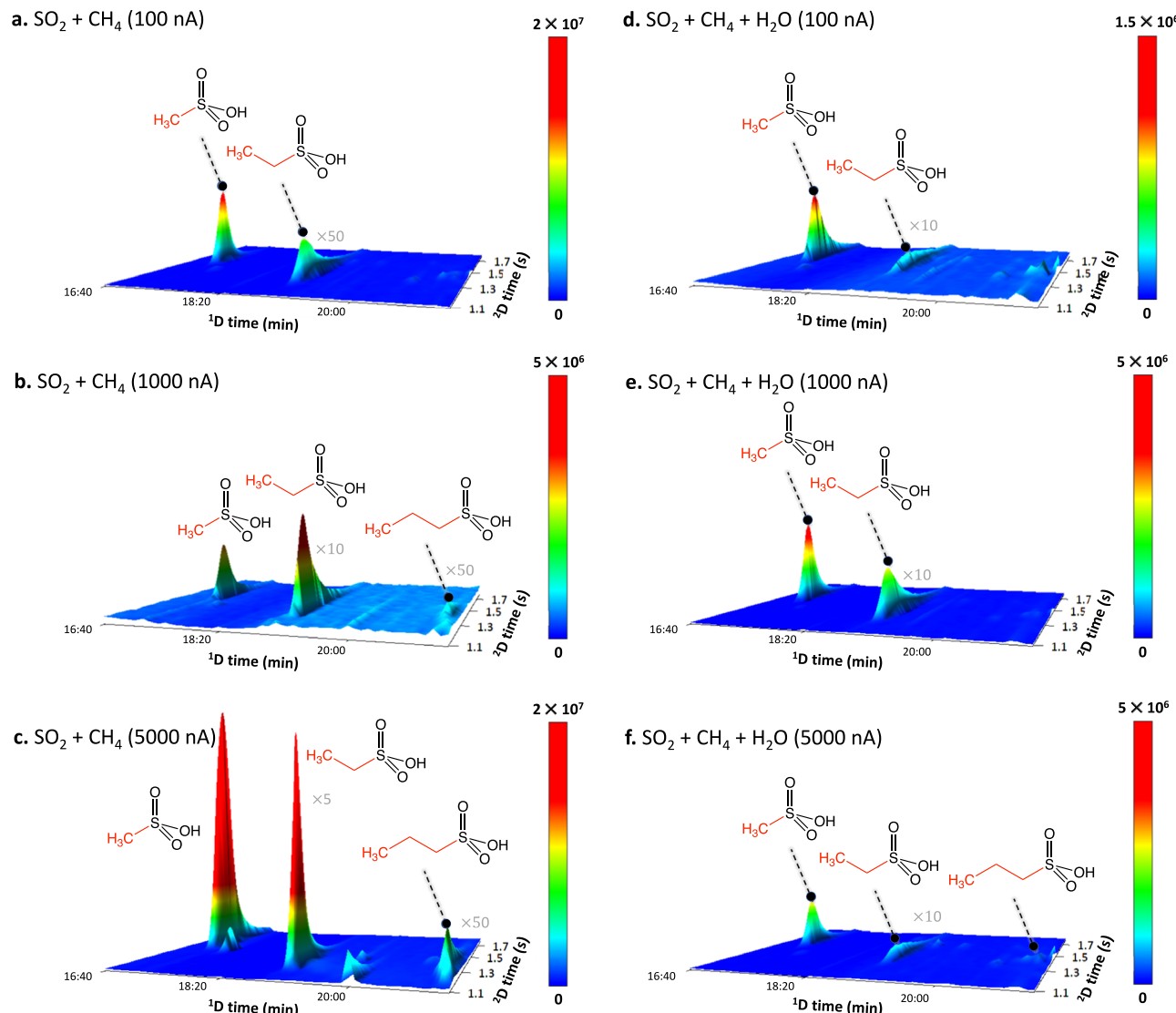

**Fig. 4 | Alkylsulfonic acids detected in room-temperature residues of the irradiated ices by two-dimensional gas chromatography coupled to time-of-flight mass spectrometry.** Gas chromatogram analyses from the **a** 100 nA SO₂/CH₄, **b** 1000 nA SO₂/CH₄, **c** 5000 nA SO₂/CH₄, **d** 100 nA SO₂/CH₄/H₂O, **e** 1000 nA SO₂/CH₄/H₂O, and the **f** 5000 nA SO₂/CH₄/H₂O experiments are shown. The peaks in the chromatograms have been scaled for clarity.

and the $t$-butyl loss channel $m/z = 153$ (M−57⁺) in all residues obtained from exposed SO₂/CH₄ and SO₂/H₂O/CH₄ ices. Similarly, ethylsulfonic acid (ESA) was identified in each experiment through the parent ion (M⁺ = 224) and the $t$-butyl loss channel (M−57⁺ = 167). Contrary to the two simplest ASAs, $n$-propylsulfonic acid (PSA) was only observed in the medium and high doses of the SO₂/CH₄ residues and the high dose of the SO₂/H₂O/CH₄ experiment via $m/z = 238$ (M⁺) and $m/z = 181$ (M−57⁺).

The quantity of ASAs in the SO₂/CH₄ (1:3) experiments was found to be significantly larger, up to two orders of magnitude, than in the SO₂:CH₄:H₂O (1:2:10) experiments, aligning with the observed trend in the intensities of the functional groups of ASAs probed in the infrared spectra. It is essential to note, that the quantitative results should be viewed as conservative estimates, given that the ASA derivatives were found to be highly unstable and decomposed rapidly after preparation. In the high dose experiments, ASAs were quantified, revealing 0.1 nmol of MSA, 0.05 nmol ESA, and 0.001 nmol of $n$-PSA in the SO₂/CH₄/H₂O system. Conversely, the SO₂/CH₄ system contained 50 nmol of MSA, 0.5 nmol of ESA, and 0.05 nmol of $n$-PSA. The formation of ASAs in the SO₂/CH₄ ices was dose-dependent, with the low dose experiments containing the smallest portion of MSA (0.5 nmol), ESA

(0.05 nmol), and no detection of $n$-PSA. This represents a reduction by more than an order of magnitude compared to the high dose experiment. In contrast, the SO₂/CH₄/H₂O systems produced approximately the same amount of ASAs in each dose, e.g., 0.1 nmol of MSA. Considering the dilution of the limiting reagents necessary to form ASAs (SO₂ and CH₄) in the water-rich ices, the amount of ASAs detected are lower than in the SO₂/CH₄ experiments. The quantities of ASAs correspond to a MSA:ESA:$n$-PSA molar ratio of 100:50:1 and 1000:10:1 for the high dose SO₂/CH₄/H₂O and SO₂/CH₄ systems, respectively. These data suggest that more complex ASAs are produced in lower quantities; a trend also reflected in the ASAs extracted from Murchison, Tagish Lake, and Allende meteorites[14,15], with, e.g., 380 nmol g⁻¹ MSA, 190 nmol g⁻¹ ESA, 81 nmol g⁻¹ PSA detected in the Murchison meteorite. Most important, the observed ratio of MSA:ESA of 2:1 in all doses of the SO₂/CH₄/H₂O system correlates well with the ratio detected in the Murchison meteorite[14]. These findings are integral to a recent analysis of the return samples from the carbonaceous asteroid Ryugu, within the framework of the Hayabusa2 mission, which revealed the presence of alkylsulfonic acids[16]. The detection of ASAs in these residues establishes an experimental framework for understanding the widespread occurrence of alkylsulfonic acids in asteroids like Ryugu and

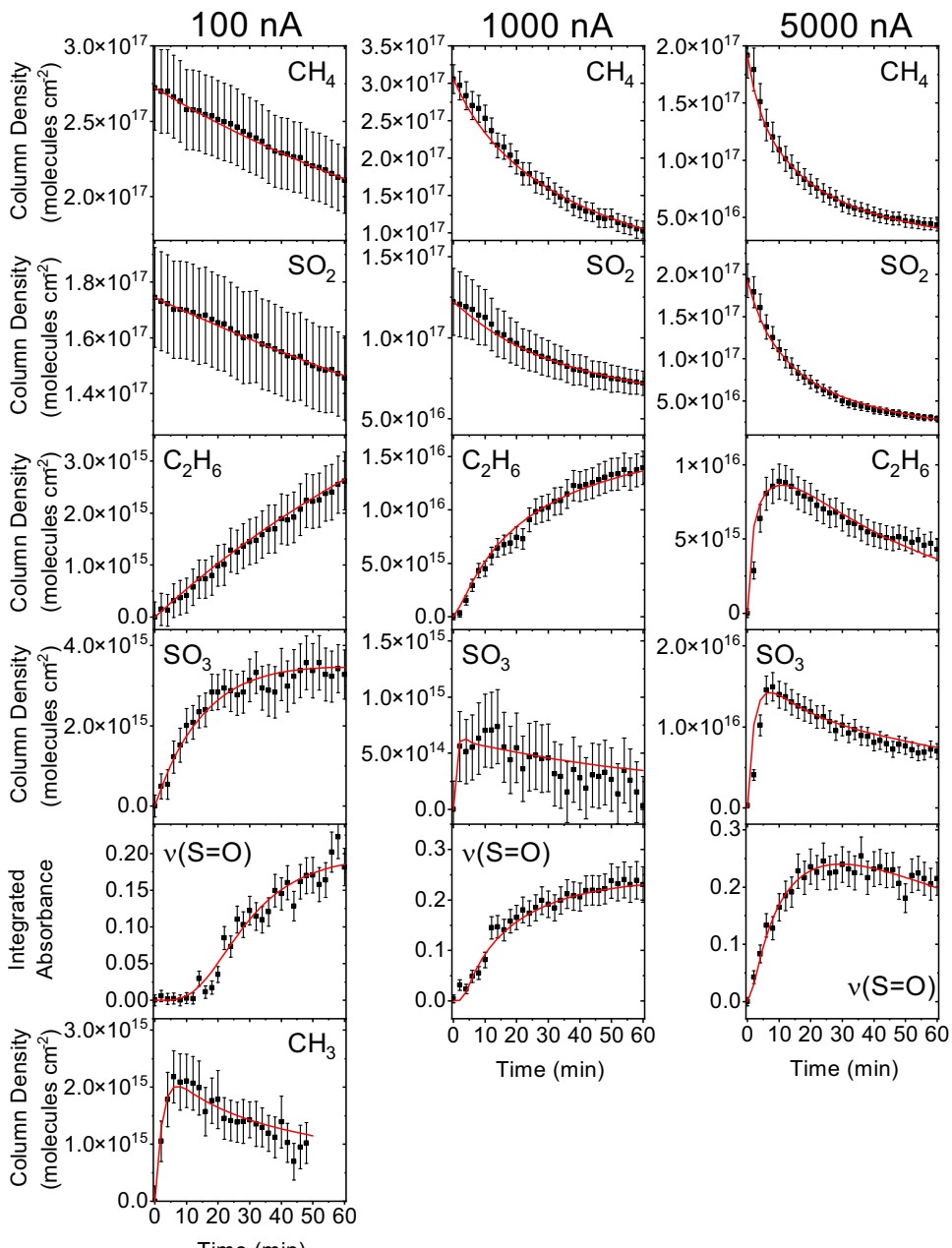

**Fig. 5 | Selected infrared peak profiles during the irradiation phase.** Each peak was tracked during the low (left), medium (center), and high (right) dose SO$_2$/CH$_4$ experiments. Methane (CH$_4$), sulfur dioxide (SO$_2$), ethane (C$_2$H$_6$), sulfur trioxide (SO$_3$), $\nu$(S=O), and the methyl radical (CH$_3^\cdot$) were tracked by the 1300 cm$^{-1}$, 1330 cm$^{-1}$, 2976 cm$^{-1}$, 1380 cm$^{-1}$, 1200 cm$^{-1}$, and 3160 cm$^{-1}$ peaks, respectively, along with their respective kinetic fits (red). The functional fits can be found in Supplementary Table 14. Error bars are calculated from instrumental noise and error in ice thickness measurements.

carbonaceous chondrites such as Murchison, Tagish Lake, and Allende. Overall, the unambiguous identification of the simplest three ASAs (MSA, ESA, n-PSA) in the residues provides compelling evidence that ASAs can be synthesized abiotically in interstellar ices and persist during the warm-up phase in hot molecular cores with overall conversion yields of MSA, ESA, and n-PSA reaching up to 5%, 0.05%, and 0.005%, respectively, with respect to the sulfur dioxide reactant.

## Discussion

Having identified the simplest three ASAs (MSA, ESA, n-PSA), we are now putting forward potential underlying formation mechanisms. These are initially inferred by comparing the molecular structures of the reactants with the ASAs, considering potential reaction intermediates, and taking into account the functional groups observed

through infrared spectroscopy (Fig. 5). Thereafter, the kinetic profiles are used to fit the proposed reaction pathways, eventually revealing the most probable formation routes of ASAs. Three conceptual routes emerge, involving sulfur dioxide (SO$_2$) (route 1), sulfur trioxide (SO$_3$) (route 2), and molecular mass growth processes from MSA via alkyl chain growth (route 3). First, the interaction of ionizing radiation with methane (CH$_4$)[39] and water (H$_2$O)[49] can induce C-H and O-H bond cleavages in overall endoergic reactions, with the necessary energy supplied by the ionizing radiation (reactions (1) and (2)). If the methyl and hydroxyl radicals are in proximity to the sulfur dioxide molecule, these radicals can undergo barrierless additon[50] (Supplementary Fig. 8) or barriered addition(1 kJ mol$^{-1}$)[51], respectively. This leads to the formation of the methylsulfonyl radical (SO$_2$(CH$_3$)) and hydroxysulfonyl radical (SO$_2$(OH)), through reactions (3) and (4),

respectively, with the reactions being exoergic by 61 and 111 kJ mol$^{-1}$. Subsequently, these doublet radicals can recombine barrierlessly with hydroxyl and methyl radicals, respectively, to form MSA through reactions (5) and (6). The overall process of synthesizing MSA from individual methane, water, and sulfur dioxide reactants is endoergic by 401 kJ mol$^{-1}$ and, hence, cannot proceed thermally at 10 K without initiation by proxies to GCRs. This intricate reaction sequence relies on the availability of reactive hydroxyl radicals and, therefore, operates in samples with abundant water content.

$$CH_4 \rightarrow CH_3 + H \quad \Delta_R G = +428 \text{ kJ mol}^{-1} \tag{1}$$

$$H_2O \rightarrow HO + H \quad \Delta_R G = +492 \text{ kJ mol}^{-1} \tag{2}$$

$$SO_2 + CH_3 \rightarrow SO_2(CH_3) \quad \Delta_R G = -61 \text{ kJ mol}^{-1} \tag{3}$$

$$SO_2 + OH \rightarrow SO_2OH \quad \Delta_R G = -111 \text{ kJ mol}^{-1} \tag{4}$$

$$SO_2 + (CH_3) + OH \rightarrow CH_3SO_2(OH) \quad \Delta_R G = -351 \text{ kJ mol}^{-1} \tag{5}$$

$$SO_2(OH) + CH_3 \rightarrow CH_3SO_2(OH) \quad \Delta_R G = -408 \text{ kJ mol}^{-1} \tag{6}$$

$$CH_4 + H_2O + SO_2 \rightarrow CH_3SO_2(OH) + 2H \quad \Delta_R G = +401 \text{ kJ mol}^{-1} \tag{7}$$

Second, sulfur dioxide (SO$_2$) can undergo radiolysis, resulting in the formation of sulfur monoxide (SO) along with ground ($^3$P) and/or excited state atomic oxygen ($^1$D) (reaction (8)). The oxygen atoms, generated with kinetic energies of a few 100 kJ mol$^{-1}$, are considered suprathermal[36] as they are not in thermal equilibrium with the surrounding 10 K ice. This excess energy allows them to overcome the reaction barrier in the subsequent formation of sulfur trioxide (SO$_3$) either barrierlessly (O($^1$D)) or with a barrier of 7 kJ mol$^{-1}$ (O($^3$P)) through reaction (9)[52]. Given that atomic oxygen possesses a triplet ground state while sulfur trioxide is a closed-shell singlet molecule, non-adiabatic dynamics must operate in this reaction. Sulfur trioxide can then react with atomic hydrogen with a 44 kJ mol$^{-1}$ barrier[53] or the methyl radical characterized by a 1 kJ mol$^{-1}$ barrier[54], forming the hydroxysulfonyl radical (SO$_2$(OH)) and methylsulfonyloxyl radical (CH$_3$SO$_3$), respectively (reactions (10) and (11)) (Supplementary Information). Barrierless recombination with a methyl radical (reaction (6)) or atomic hydrogen (reaction (12)) results in the formation of MSA. It is important to note that the atomic oxygen, which is required to form sulfur trioxide via reaction (9), is not only supplied through the radiolysis of sulfur dioxide, but also through the decomposition of water (reaction (13))[55]. This feature enables the realization of this complex reaction sequence in both polar (water-rich) and non-polar (water-depleted) ices.

$$SO_2 \rightarrow SO + O(^3P/^1D) \quad \Delta_R G = +547/+737 \text{ kJ mol}^{-1} \tag{8}$$

$$SO_2 + O(^3P/^1D) \rightarrow SO_3 \quad \Delta_R G = -342/-532 \text{ kJ mol}^{-1} \tag{9}$$

$$SO_3 + H(^2S) \rightarrow SO_2(OH) \quad \Delta_R G = -214 \text{ kJ mol}^{-1} \tag{10}$$

$$SO_3 + CH_3 \rightarrow CH_3SO_3 \quad \Delta_R G = -64 \text{ kJ mol}^{-1} \tag{11}$$

$$CH_3SO_3 + H \rightarrow CH_3SO_2(OH) \quad \Delta_R G = -474 \text{ kJ mol}^{-1} \tag{12}$$

$$H_2O \rightarrow O(^1D) + H_2 \quad \Delta_R G = +676 \text{ kJ mol}^{-1} \tag{13}$$

Finally, MSA could facilitate molecular mass growth processes through the elongation of the alkyl chain. In this route, the radiolytic decomposition of ASA via hydrogen atom loss from the methyl group gives rise to a hydroxysulfonylmethyl radical (CH$_2$SO$_2$(OH)) (reaction (14)). Successive recombination of the latter with alkyl radicals such as methyl (CH$_3$) and even ethyl (C$_2$H$_5$) enables the formation of ESA and PSA, respectively (reactions (15) and (16)) via barrierless radical-radical recombination reactions.

$$CH_3SO_2(OH) \rightarrow CH_2SO_2(OH) + H \tag{14}$$

$$CH_2SO_2(OH) + CH_3 \rightarrow CH_3CH_2SO_2(OH) + H \tag{15}$$

$$CH_2SO_2(OH) + C_2H_5 \rightarrow CH_3CH_2CH_2SO_2(OH) + H \tag{16}$$

What are the actual pathways leading to ASAs? It is essential to emphasize that, unlike crossed molecular beam experiments that explore the formation of organosulfur molecules in bimolecular gas-phase reactions at the molecular level[56–59], the condensed-phase environment in the present experiments represents a complex environment. This complexity results from the collisional stabilization of reaction intermediates, the transfer of their internal energy to the surrounding matrix (ice), and successive reaction sequences involving branched and consecutive reactions. Therefore, the elucidation of the underlying reaction mechanisms relies on the identification of reaction intermediates (SO$_3$, CH$_3$, C$_2$H$_6$) along with newly emerging functional groups during the radiolysis of the ices at 10 K. Likely reaction pathways are further supported by the temporal evolution of the infrared bands of the reactants, products, and precursors. This evolution was monitored over the irradiation time at 10 K, particularly for systems with sufficient signal-to-noise ratios in the newly emerging infrared absorptions. This comprehensive analysis has been accomplished for the SO$_2$/CH$_4$ system. Figure 5 reports the temporal evolution of the reactants (CH$_4$, SO$_2$), intermediates (SO$_3$, CH$_3$, C$_2$H$_6$), and functional groups of ASAs (S=O). Additionally, the kinetic scheme exploited to fit the experimental data of the CH$_4$/SO$_2$ system is depicted, and corresponding rate constants are compiled in Supplementary Table 14 (Supplementary Information). These kinetic fits were exploited to inform a possible reaction pathway to generate ASAs. Specifically, the delay in the production of ASAs—as observed in the 100 and 1000 nA experiments—suggest multiple intermediates are crucial to its formation. We would expect reactions involving reagent material, such as SO$_2$, to show rapid production due to the orders of magnitude difference in availability compared to intermediates like SO$_3$. Here we find swift synthesis of SO$_3$, CH$_3$, and ethane (C$_2$H$_6$) but a delay in production of ASAs suggesting compounds necessary to form ASAs are not present at the start of the irradiation. Based on these fits, the synthesis of ASAs predominantly occurs through the addition of a methyl radical to sulfur trioxide (reaction (11)); the formation of sulfur trioxide via reactions (8) and (9) is critical for synthesizing ASAs in these ices. Finally, recombination of the methylsulfonyloxyl radical (CH$_3$SO$_3$) with atomic hydrogen (reaction (12)) lead to the formation of ASAs. Notably, water is not required to produce these organosulfur compounds in this system. However, the hydroxyl radical—supplied from irradiated water—may allow an alternate pathway towards ASAs as shown via reactions (1)–(6) in water-rich ices. Alternatively, reactions leading to ASAs and their precursors could form from other chemical species. Material such as hydrogen sulfide presents a promising starting point, whereby successive oxidation by water or carbon dioxide (CO$_2$) in methane-containing ices could produce ASAs. The possibility of higher-order ASAs emerging through molecular mass growth processes is

suggested, whereby hydrogen loss and recombination with an alkyl radical leads to higher mass products, resembling the formation of polycyclic aromatic hydrocarbons (PAH) in the ISM[60]. The detection of $C_{14}$-length ASAs in meteoritic material[15] and $C_7$-length chains in samples returned from Ryugu[16] suggests that molecular mass growth processes are an integral part in producing these larger homologs.

Our study affords persuasive evidence for a facile synthesis of the three simplest alkylsulfonic acids (methylsulfonic acid (MSA, $CH_3SO_2(OH)$), ethylsulfonic acid (ESA, $CH_3CH_2SO_2(OH)$), propylsulfonic acid (PSA, $C_3H_7SO_2(OH)$)). This synthesis occurs upon exposing sulfur dioxide-bearing interstellar analog ices to proxies of GCRs over a lifespan equivalent to that of cold molecular clouds reaching $5 \times 10^7$ years. The identification of these S(+IV) compounds provides compelling affirmation that alkylsulfonic acids as detected in sources like the Murchison meteorite and recently in return samples of the carbonaceous asteroid Ryugu, represent a potential source of bioavailable sulfur for the organic feedstock driving prebiotic evolution. These acids can be synthesized abiotically within the icy mantles of interstellar grains present in molecular clouds. The identification of alkylsulfonic acids further highlights the presence of S(+IV) and the S-O-C backbone, which is omnipresent in contemporary biomolecules. Considering the facile preparation of the S(+IV)-O-C moiety in interstellar ice analogs, introducing sulfur dioxide, hydrogen sulfide, and carbon disulfide to these ices could yield fundamental, yet more complex building blocks of biomolecules such as sulfoquinovose ($C_6H_{12}O_8S$), 3′-phosphoadenosine-5′-phosphosulfate ($C_{10}H_{15}N_5O_{13}P_2S$), coenzyme M ($HSCH_2CH_2SO_2(OH)$), and taurine ($H_2NCH_2CH_2SO_2(OH)$). The understanding of the synthesis of ASAs in interstellar analog ices marks the very first step toward solving the interstellar sulfur problem. Previous studies have found that sulfur levels in dense molecular clouds and circumstellar regions around young stellar objects are significantly lower than the cosmic abundance by up to three orders of magnitude leading to the so-called "sulfur depletion problem"[61,62]. The presence of ASAs in these residues indicates that sulfur might remain trapped as organosulfur species on the surface of interstellar grains in these dense regions, thus depleting molecular clouds of sulfur in the gas phase. The suite of extraterrestrial sulfur-containing molecules is expanding rapidly, as evidenced by the recent identification of hydroxy alkylsulfonic acids (HO-R-$SO_2(OH)$), alkylthio-sulfonic acids (R-S(S)(O)OH), and hydroxy alkylthiosulfonic acids (HO-R-S(S)(O)OH) in samples returned from Ryugu[16]. This finding demonstrates the pressing need to rationalize the previously marginalized sulfur chemistry in astrophysical environments. Laboratory astrophysics research aims to enhance knowledge of astrophysical systems. Our results bestow fundamental knowledge toward the abiotic sulfur chemistry in extraterrestrial environments, extending beyond interstellar ices to encompass meteorites and even comets such as 67 P/Churyumov-Gerasimenko, where molecules with the molecular formulae like $C_2H_6OS$, $CH_3O_2S$, $CH_3S_2$, and $CH_2NS$ have been identified[63]. Considering that molecular clouds are nurseries of stars and planetary systems, the detection of alkylsulfonic acids proposes that these molecules could have been at least partially delivered to our Solar System and others from their interstellar nurseries via circumstellar disks. These compounds subsequently became incorporated in asteroids and comets prior to their delivery to Earth. This process forces us to reconsider the inventory and complexity of sulfur-bearing molecular precursors available on early Earth, which are relevant to the Origins of Life.

## Methods

### Surface scattering machine

Experiments were performed in a stainless steel ultrahigh-vacuum chamber held at $8 \times 10^{-11}$ Torr by a magnetically levitated turbomolecular pump (Osaka, TG420MCAC) backed by an oil-free dry scroll pump (Edwards, GVSP30)[39,64]. A mirror-polished silver substrate was attached to a cold finger cooled to $10.0 \pm 0.3$ K by a closed-cycle helium compressor (Helix, 9600 Compressor). Two sets of

experiments were conducted. First, methane ($CH_4$, Sigma Aldrich, 99.998%) and sulfur dioxide ($SO_2$, Matheson, 99.98%) gases were premixed in a 3:1 ratio in a gas mixing chamber and deposited at a pressure of $3 \times 10^{-7}$ Torr via a 2.5 cm diameter glass capillary array onto the cooled substrate. Second, a tertiary ice mixture of water ($H_2O$) with $SO_2$ and $CH_4$ was prepared using HPLC grade $H_2O$ that was degassed with three freeze-pump-thaw cycles. The premixed $CH_4/SO_2$ gas (3:1) was deposited at a pressure of $2 \times 10^{-7}$ Torr while $H_2O$ was simultaneously deposited utilizing a separate glass capillary array at $3 \times 10^{-8}$ Torr. Isotopically labeled ices were also prepared using the same gas mixture ratio with methane-$d_4$ ($CD_4$, Sigma Aldrich, 99.9 atom % D) and methane-$^{13}$C ($^{13}CH_4$, Sigma Aldrich, 99 atom % $^{13}$C).

The gas deposition was monitored utilizing interferometry by a HeNe laser (Melles Griot, 25-LHP-230, 632.8 nm) at an angle of incidence of 3°[65]. The thickness of the ice was calculated based on the refractive index ($n$) of amorphous methane ($n = 1.30 \pm 0.02$[66], sulfur dioxide ($n = 1.36$)[67], and water ($n = 1.27 \pm 0.02$)[68]. A weighted average based on the ice composition was exploited ($SO_2$:$CH_4$, 2:5, $n = 1.32 \pm 0.02$ and $SO_2$:$CH_4$:$H_2O$, 1:2:10, $n = 1.28 \pm 0.03$). The ice thicknesses of the $SO_2/CH_4$ and $SO_2/CH_4/H_2O$ ices were calculated to be $1,050 \pm 50$ nm and $950 \pm 40$ nm, respectively. This is thicker than the 95th percentile penetration depth of the electrons ($890 \pm 60$ nm, $SO_2$/$CH_4$ and $750 \pm 70$ nm, $SO_2/CH_4/H_2O$) determined by CASINO Simulations[69].

The relative fractions of the reactant molecules in the ices was determined using a Fourier Transform infrared (FTIR) spectrometer (Thermo, Nicolet 6700) covering the 6000 to 500 cm$^{-1}$ range at a spectral resolution of 4 cm$^{-1}$ utilizing a liquid nitrogen cooled HgCdTe (Thermo, MCT-B) detector. The relative concentrations of reactants in the deposited ices were computed using integrated infrared absorption coefficients of $H_2O$ ($v_2$, 1660 cm$^{-1}$, $1.2 \times 10^{-17}$ cm molecule$^{-1}$; $v_1/v_3$, 3280 cm$^{-1}$, $1.4 \times 10^{-16}$ cm molecule$^{-1}$)[46,70], $SO_2$ ($v_1$, 1149 cm$^{-1}$, $2.2 \times 10^{-18}$ cm molecule$^{-1}$; $v_3$, 1335 cm$^{-1}$, $1.47 \times 10^{-17}$ cm molecule$^{-1}$)[29], and $CH_4$ ($v_3 + v_4$, 4301 cm$^{-1}$, $5.6 \times 10^{-19}$ cm molecule$^{-1}$; $v_1 + v_4$, 4202 cm$^{-1}$, $3.00 \times 10^{-19}$ cm molecule$^{-1}$)[71]. The $SO_2$:$CH_4$ and $SO_2$:$CH_4$:$H_2O$ ices ratios were determined to be 2:5 ± 2 and 1:2:10 ± 1, respectively.

After the deposition, each sample was isothermally irradiated at 10 K with 5 keV electrons generated by an electron gun (Specs EQ 22-35) at beam currents of 100 nA (low dose), 1000 nA (medium dose), and 5000 nA (high dose) for 60 minutes at an angle of 15°. According to CASINO simulations, the high dose experiment corresponds to doses $170 \pm 30$ eV molecule$^{-1}$ and $57 \pm 9$ eV molecule$^{-1}$ for $SO_2$ and $CH_4$, respectively, in the $SO_2/CH_4$ experiments and $210 \pm 30$ eV molecule$^{-1}$, $70 \pm 11$ eV molecule$^{-1}$, and $66 \pm 11$ eV molecule$^{-1}$ for $SO_2$, $CH_4$, and $H_2O$, respectively, in the $SO_2/CH_4/H_2O$ ices. These doses scale linearly with current and the calculations use the pure ice with a thickness of 1 μm, and incorporate the corresponding mixed ice density. The density of each ice component ($SO_2$, 1.395 g cm$^{-3}$[72]; $CH_4$, 0.47 g cm$^{-3}$[66]; $H_2O$, 0.97 g cm$^{-3}$[73]) in their respective ratios in the ice were utilized to calculate the final mixed ice density ($SO_2$:$CH_4$, 2:5 ± 2, $0.73 \pm 0.05$ g cm$^{-3}$; $SO_2/CH_4/H_2O$, 1:2:10 ± 1, $0.90 \pm 0.01$ g cm$^{-3}$). FTIR spectra of the ices were collected before, during, and after the irradiation to monitor the emergence of new functional groups by averaging spectra over 2 minutes. The error in FTIR data found in Fig. 5 and Supplementary Fig. 7 was calculated using error propagation that incorporated one standard deviation of the FTIR noise and the errors in the angle of incidence, ice density, refractive indices, and absorption coefficients utilized in the ice thickness and column density calculations.

After the irradiation, the ices were heated using temperature-programmed desorption (TPD) from 10 K to 320 K at a rate of 1 K min$^{-1}$ by programmable temperature controller (Lakeshore 336). During TPD, FTIR spectra were once again averaged over 2 minute intervals to monitor thermal and chemical changes, while molecules subliming from the surface were analyzed by an electron-impact ionization

quadrupole mass spectrometer (QMS, Balzer, QMG 420) with an electron energy of 100 eV and emission current of 1 mA.

## Two-dimensional gas chromatography time of flight mass spectrometry

Solid residues that remained after the TPD phase were stored under dry conditions and then extracted ex situ with $10 \times 50 \mu L$ of Milli-Q water (Milli-Q Direct 8, 18.2 MΩ cm at 298 K, <2 ppb total organic carbon) and transferred into conical reaction vials (1 mL, V-Vial Wheaton). The solutions were dried under a gentle stream of dry nitrogen. Subsequently, 20 μL of $10^{-6}$ M methyl laurate (internal standard, Sigma Aldrich, purity ≥ 97.0%) in hexane (Sigma Aldrich, ≥ 99.0%) was added to the reaction vials and the mixtures were dried. Finally, the residues were dissolved in 20 μL of *N-tert*-butyldimethylsilyl-*N*-methyltrifluoroacetamide (MTBSTFA, Sigma Aldrich, ≥99.0%) by vigorously stirring the reaction mixtures for 1 min using a vortex (Stuart Scientific, SA8), and transferred to 1 mL GC vials equipped with 100 μL inserts (Agilent Technologies). All glassware used during the analytical procedure was flushed 12 times with Milli-Q water, 12 times with ethanol (TechniSolv, 96%), 12 times with Milli-Q water, and subsequently heated at 773 K for 5 h. Eppendorf pipette tips, GC vials, inserts, and caps were used without further cleaning.

The analyses were carried out using two-dimensional gas chromatography (GC×GC, Pegasus BT 4D) coupled to electron-ionizition reflectron time-of-flight mass spectrometry (TOF-MS, LECO Corp.)[17,74]. The TOF-MS system operated at a storage rate of 150 Hz, with a $40-500 \ m/z$ range, and a solvent delay of 16.4 min. The injector and ion source temperatures were kept at 503 K and the transfer line at 513 K. The column set consisted of a DB-5MS Ultra Inert primary capillary column (29.72 m × 0.25 μm, Agilent) press-tight (Restek Corp.) connected to a DB Wax (polyethylene glycol, 1.11 m × 0.1 mm, 0.1 μm, Agilent) secondary column. Modulation between the columns was ensured via a liquid nitrogen jet-based thermal modulator. Hydrogen was used as carrier gas at a constant flow rate of 1.5 mL min⁻¹. The primary oven utilized a temperature gradient starting at 313 K for 1 min, then increased at a rate of 5 K min⁻¹ to 488 K, where it was held for 5 min. The secondary oven operated with a constant positive temperature offset of 283 K. The thermal modulator hot jets temperature offset was set at 288 K with a modulation period of 5 s. Aliquots of 1 μL were injected in splitless mode. The data was processed using Leco corp. ChromaTOF® software.

## Data availability

All data generated in this study is available in the main text and the supplementary materials. Source data are provided with this paper.

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

## Acknowledgements

The authors would like to thank the W. M. Keck Foundation (R.I.K.) and the University of Hawaii (M.E.M., A.H.) for supporting the surface science machine. The Hawaii team thanks the US National Science Foundation (NSF) Division for Astronomy (NSF-AST 2103269) for support. Further funding is acknowledged by the European Research Council through the European Union's Horizon 2020 research and innovation program, under grant agreement 804144 (C.M.) as well as the National Centre for Space Studies (CNES) for a postdoctoral fellowship (J.B.).

## Author contributions

M.E.M., A.H., A.M.T. carried out the experiments; J.B. and C.M. performed gas chromatography analysis; M.E.M. and J.B. analyzed the experimental data; M.E.M., A.M.T., and R.I.K. wrote the manuscript; and R.I.K. supervised the study.

## Competing interests

The authors declare no competing interests.
