## [Peer Review File · Nature Communications]

Abiotic formation of alkylsulfonic acids in interstellar analog ices and implications for their detection on RyuguREVIEWER COMMENTS

Reviewer #1 (Remarks to the Author):

The manuscript by McAnally and Bockova et al., presents a study of the formation of alkylsulfonic acids in interstellar ice analogues consisting of SO₂, CH₄, and H₂O. The authors study the irradiation of ice films in-situ with IR spectroscopy, identify newly formed functional groups, and study their kinetics. The alkylsulfonic acids are detected ex-situ by employing GCxGC-MS. In general, the study is thorough, and the manuscript reads well. Consequently, I only have a small number of comments which I have grouped into MAJOR (which sounds more daunting than it is) and MINOR (which are mostly small clarifications).

MAJOR

p4 line 65-68: I have a couple of issues with this sentence: 1) it is Herschel/HIFI, 2) in the context of SO₂ on grains, I'm not sure why the gas-phase observations of Danilovich et al., are included and would suggest omitting this reference, 3) while SO₂ is certainly a prominent source of sulfur, it also only accounts for just a few % of the S budget, see Boogert+2015, and I think it is important to acknowledge and discuss this, 4) there are some older references that are worth including, such as Boogert+1997, or Rosetta results of comet 67P if the authors want to link to the Solar System.

P7, line 159-161, line 180-181, and methods section: The SO₂:CH₄:H₂O is only given in line 180-181. Can the authors also give this ratio at other relevant locations, such as the methods section (for the refractive index calculation) and in line 159-161? In the latter case, the authors state that ASA-associated IR bands are an order of magnitude lower in the SO₂:CH₄:H₂O experiments than in the SO₂:CH₄ experiments and this is consistent with the lower quantities of SO₂ and CH₄. Naively, I expect a deposition ratio of (SO₂:CH₄):H₂O of 1:10, which is not the case (it is about 1:3). Instead of the lower availability of reagents, is the 1-2 orders of magnitude lower product abundance not more likely to be the result of competition of chemical reactions with H₂O and its radiolysis products? Can add this perspective to the manuscript or present reasoning as to why they disagree.

P7, line 164-165: The Kaiser group is of course known for their PI-ReToF-MS technique, which is not used in this study. Why is that? At first, I suspected that the sublimation temperature of ASAs is too high for the TPD, but when I checked the boiling point of methylsulfonic acid and ethylsulfonic acid, I found they are lower than that of N-methylformamide – a molecule that has been studied using PI-ReToF-MS (Frigge+2018). This might mean that MSA and ESA do desorb from the surface before the final TPD temperature (320 K). Can the authors comment on this and elaborate on why ASAs will not desorb in a vacuum or only partially do so? It might be useful to include this information in the manuscript.

P11, line 285-287: I'm not sure if follow or agree with the reasoning of the authors. It seems that the authors prefer the SO₃ + CH₃ reaction, because both these reactants are rapidly formed at early irradiation time scales (correct?). However, to me, this doesn't make reaction [11] more prominent than the barrierless reaction SO₂ + CH₃ and subsequent reactions, especially considering the larger abundance of SO₂. Can the authors present more elaborate reasoning in the manuscript as to why they indicate reactions involving SO₃ as the dominant ones to produce ASAs?

MINOR

p3. line 38: the "where R represents an alkyl group" should be placed within the brackets.

p3 line 40: the way the sentence "utilized by methanobacteria and taurine" is phrased gives the impression that taurine also is an organism.

p4 line 80-81: For completeness of information, can the authors list which doses they use in each of the three cases? (or, alternatively, a dose rate)

fig 2, caption: it might be useful to mention the meaning of the dashed line and the 5x/10x sign.

Fig 3, caption: this is the FTIR spectrum of the residue of SO₂/CH₄, not the ice itself.

P5, line 114-119: While the second sentence claims the formation of functional groups of ASAs, the first sentence seems to suggest that ASAs are identified. I would suggest that the authors in sentence one already state that functional groups linked to ASAs are identified.

P5, line 121-123: similar to the above point, this sentence suggests that ASAs are detected by their IR signatures, while only IR features indicative of functional groups linked to ASAs are present. I'd suggest that the authors look at the phrasing of this and other sentences to clarify that IR features of functional groups that can be linked to ASAs are present, but not that ASAs are directly detected.

P6, line 136: The authors claim that 22+/-9% of the functional groups remain after warm-up from 10 to 320 K, which I read as: the remaining ~78% has sublimated. IR absorption bands often change (that is, can decrease) due to heating and rearrangement of the molecular structures, which can therefore also account for the decrease in absorbance. I would suggest omitting the statement about the remaining % to avoid confusion about this point and simply state that IR signatures of ASAs functional groups are still present at 320K.

P6, line 144: A comparative statement is made about the experiments with and without H₂O. I do not understand why the Hagen et al., 1981 paper is cited here and suggest to remove it.
Fig S6: In the caption, the green fit line for reagents is described, but the green fit is not used in the figure. I would therefore remove it from this caption. Please also check other figures for similar instances.

P9, line 221-222: "these radicals can undergo addition to the sulfur atom via barrierless addition with a barrier of 1 kJ mol⁻¹ (Fig. S8)". Barrierless addition with a barrier seems like a contradiction and requires further explanation. Furthermore, I cannot identify the 1 kJ mol⁻¹ barrier in Fig. S8.

P14, line 375-378: Can the authors provide more details on how the dose is calculated? Specifically, are these doses determined for pure or mixed ices, and was this for the ~1μm thick ice film? If these are calculations for pure ice, how does the dose change in mixed ice?

Reviewer #2 (Remarks to the Author):

The study by Mason McAnally et al. explores the abiotic formation of alkylsulfonic acids in interstellar sulfur-doped ices, which are believed to have played a crucial role in early Earth biochemistry. Laboratory simulations demonstrated that these acids can be efficiently produced through the interaction of sulfur dioxide, water, and methane ices with galactic cosmic rays, mimicking conditions in cold molecular clouds. This finding has significant astrophysical implications, suggesting that alkylsulfonic acids could have been synthesized in interstellar environments and delivered to early Earth via comets and asteroids, providing a soluble source of organosulfur compounds essential for the development of prebiotic chemistry. The identification of these acids in meteorites and the carbonaceous asteroid Ryugu supports this hypothesis, highlighting their potential role in the origins of life and the sulfur chemistry in extraterrestrial environments.

The article is very well written and the experimental results is well presented and discussed. I recommend publication after the authors address the following comments:

My main comments are:

1) A discussion about eventual contamination during ex-situ GC-MS analysis and sample holding is needed.

How were the samples stored at room temperature and for how long?

2) A discussion about the timescale on astrophysical environments concerning the employed electron beam is needed.

3) Authors should also clarify the readers about the link between the employed electron beam and cosmic rays (with a discussion also about the different penetration depth of both projectiles within matter, and eventual different physicochemical products)

4) Several years ago, Professor Pilling's group conducted studies using X-rays on ices containing SO₂ (Pilling & Bergantini, 2015, *The Astrophysical Journal*, 811(2), 151; Bonfim et al., 2017, *Phys. Chem. Chem. Phys.*, 19, 26906-26917) and the implantation of S ions in ices (Boduch et al., 2015, *Journal of Physics: Conference Series*, 629, 012008; Lv et al., 2014, *Mon. Not. R. Astron. Soc.*, 438, 922-929). These studies could provide valuable insights into the processing of SO₂-rich ices due to ionizing radiation. The authors may consider citing these studies in their current publication to clarify this aspect for the reader.

5) Authors should add some lines about the possibility of others reaction routes in the formation of the important Alkylsulfonic acids and its precursors.

6) What are the red lines in figure 5? Models?

Reviewer #3 (Remarks to the Author):

This paper is generally well written and the results are well presented. A pdf is provided with suggestions for improvement.

Reviewer #1 (Remarks to the Author):

The manuscript by McAnally and Bockova et al., presents a study of the formation of alkylsulfonic acids in interstellar ice analogues consisting of SO₂, CH₄, and H₂O. The authors study the irradiation of ice films in-situ with IR spectroscopy, identify newly formed functional groups, and study their kinetics. The alkylsulfonic acids are detected ex-situ by employing GCxGC-MS. In general, the study is thorough, and the manuscript reads well. Consequently, I only have a small number of comments which I have grouped into MAJOR (which sounds more daunting than it is) and MINOR (which are mostly small clarifications).

MAJOR

p4 line 65-68: I have a couple of issues with this sentence: 1) it is Herschel/HIFI, 2) in the context of SO₂ on grains, I'm not sure why the gas-phase observations of Danilovich et al., are included and would suggest omitting this reference, 3) while SO₂ is certainly a prominent source of sulfur, it also only accounts for just a few % of the S budget, see Boogert+2015, and I think it is important to acknowledge and discuss this, 4) there are some older references that are worth including, such Boogert+1997, or Rosetta results of comet 67P if the authors want to link to the Solar System.

Reply: We have removed the reference and opted to include the connection to comets such as 67P (Calmonte+2016). We have added a discussion of the various sulfur-containing molecules found in the ISM.

“Sulfur dioxide (SO₂) presents a promising source of sulfur in interstellar grains and comets considering recent detections in the coma of 67P/Churyumov-Gerasimenko²³ and by the James Webb Space Telescope²⁴ at levels of 0.05% compared to water (H₂O). Although SO₂ is a prominent source of sulfur in the ISM, it only contributes to a few percent of the total sulfur budget²⁵. The remaining sulfur is characterized in several different forms including iron(II) sulfide (FeS), magnesium sulfide (MgS), carbonyl sulfide (OCS), and hydrogen sulfide (H₂S)²⁶.”

P7, line 159-161, line 180-181, and methods section: The SO₂:CH₄:H₂O is only given in line 180-181. Can the authors also give this ratio at other relevant locations, such as the methods section (for the refractive index calculation) and in line 159-161?

Reply: We have added the ice ratio to lines 109, 159-161, and to the methods section.

These results clearly reveal that functional groups of ASAs are also synthesized in the SO₂:CH₄:H₂O (1:2:10) ices...”

“The infrared spectra of the deposited sulfur dioxide/methane (SO₂:CH₄, 2:5) ices...”

And in the methods section: “A weighted average based on the ice composition was exploited (SO₂:CH₄, 2:5, $n = 1.32 \pm 0.02$ and SO₂:CH₄:H₂O, 1:2:10, $n = 1.28 \pm 0.03$).”

In the latter case, the authors state that ASA-associated IR bands are an order of magnitude lower in the SO₂:CH₄:H₂O experiments than in the SO₂:CH₄ experiments and this is consistent with the lower quantities of SO₂ and CH₄. Naively, I expect a deposition ratio of (SO₂:CH₄):H₂O of 1:10, which is not the case (it is about 1:3). Instead of the lower availability of reagents, is the 1-2 orders of magnitude lower product abundance not more likely to be the result of competition of chemical reactions with H₂O and its radiolysis products? Can add this perspective to the manuscript or present reasoning as to why they disagree.

Reply: This is correct. Two factors limit the production of ASAs in the water-rich ices.

1) Limitation of reagents necessary to produce ASAs (SO₂ and CH₄).

2) Competing reactions involving H₂O and its radiolysis products limiting the ability for ASA to form

We have added clarification of this (line 163-164). This section now states:

“These results clearly reveal that functional groups of ASAs are also synthesized in the SO₂:CH₄:H₂O (1:2:10) ices upon irradiation and warmup; however, the intensity of peaks likely associated with ASAs are an order of magnitude lower in the water-rich ice, which is consistent with lower quantities of limiting reagents—SO₂ and CH₄—required to produce ASAs. Additionally, radiolysis products derived from water may inhibit the formation of ASAs due to competing reactions consuming crucial intermediates.”

P7, line 164-165: The Kaiser group is of course known for their PI-ReToF-MS technique, which is not used in this study. Why is that? At first, I suspected that the sublimation temperature of ASAs is too high for the TPD, but when I checked the boiling point of methylsulfonic acid and ethylsulfonic acid, I found they are lower than that of N-methylformamide – a molecule that has been studied using PI-ReToF-MS (Frigge+2018). This might mean that MSA and ESA do desorb from the surface before the final TPD temperature (320 K). Can the authors comment on this and elaborate on why ASAs will not desorb in a vacuum or only partially do so? It might be useful to include this information in the manuscript.

Reply: Yes, the referee is correct that our group has the PI-ReToF-MS technique. This technique depends on the availability of adiabatic ionization energies (IEs) of ALL isomers of, e.g., methylsulfonic acid plus their conformers. Further, of ranges of IEs of one the isomers overlap, UVVIS spectra have to be computed to selectively photodissociate one isomer, but not the other. This procedure has to be conducted for ethyl, n/i propyl ASAs, too. Therefore, PI-ReToF-MS experiments are not always the most economical or optimal choice. Consequently, given our focus on the homologous series of alkylsulfonic acids, which can be effectively separated using chromatographic methods, we decided for the GCxGC-MS approach in this study.

P11, line 285-287: I'm not sure if follow or agree with the reasoning of the authors. It seems that the authors prefer the $\text{SO}_3 + \text{CH}_3$ reaction, because both these reactants are rapidly formed at early irradiation time scales (correct?). However, to me, this doesn't make reaction [11] more prominent than the barrierless reaction $\text{SO}_2 + \text{CH}_3$ and subsequent reactions, especially considering the larger abundance of SO_2 . Can the authors present more elaborate reasoning in the manuscript as to why they indicate reactions involving SO_3 as the dominant ones to produce ASAs?

Reply: The kinetic fits inform our selection for choosing SO_3 and CH_3 as the primary reaction pathway. Specifically, the delay in the production of the functional group associated with ASAs points to multiple intermediates being necessary. Typically reactions involving reagents like SO_2 have sharp increases since the material is orders of magnitude more dominant in the ice. A profile involving reagent material would be expected to look similar to the SO_3 production profile, CH_3 profile, or C_2H_6 . In the case of ASAs we find a delay before production. Typically this delay occurs because material necessary to form it is missing from the ice (or in extremely low abundances) and must be formed before the reaction can finally occur.

Secondly, the difference in barriers (or lack thereof) are based on gas-phase computations rather than condensed phase with errors around 10 kJ mol^{-1} . Thus these reactions are not dissimilar enough to pick one reaction over the other based purely on these computations.

We have added some clarification about the kinetic fits and their possible implications.

“These kinetic fits were exploited to inform a possible reaction pathway to generate ASAs. Specifically, the delay in the production of ASAs—as observed in the 100 and 1000 nA experiments—suggest multiple intermediates are crucial to its formation. We would expect reactions involving reagent material, such as SO_2 , would show rapid production due to the orders of magnitude difference in availability compared to intermediates like SO_3 and would reflect first or second order kinetics.”

MINOR

p3. line 38: the “where R represents an alkyl group” should be placed within the brackets.

Reply: Fixed.

p3 line 40: the way the sentence “utilized by methanobacteria and taurine” is phrased gives the impression that taurine also is an organism.

Reply: We have clarified this section to prevent any misinterpretation that taurine is an organism. This now reads:

“Alkylsulfonic acids (ASAs, $\text{RSO}_2(\text{OH})$ where R represents an alkyl group, Fig. 1) typifies the S(+IV) oxidation state of sulfur present in key biological systems such as coenzyme M ($\text{HSCH}_2\text{CH}_2\text{SO}_2(\text{OH})$), which is utilized by methanobacteria⁵, and taurine ($\text{H}_2\text{NCH}_2\text{CH}_2\text{SO}_2(\text{OH})$), i.e., a prominent molecular component in bile⁶ and energy source of

prokaryotes⁷.”

p4 line 80-81: For completeness of information, can the authors list which doses they use in each of the three cases? (or, alternatively, a dose rate)

Reply: We have added the dose to this section.

“Three doses were selected to represent various ages of molecular clouds: a low dose simulating 10^6 years in cold molecular clouds (up to $4.2 \text{ eV molecule}^{-1}$), a medium dose (10^7 years; up to $42 \text{ eV molecule}^{-1}$), and a high dose (5×10^7 years, up to $210 \text{ eV molecule}^{-1}$).”

fig 2, caption: it might be useful to mention the meaning of the dashed line and the 5x/10x sign.

Reply: We have clarified the meaning of the dashed line by adding:

“The spectral region ($2000\text{--}500 \text{ cm}^{-1}$) to the right of the dashed vertical line has been magnified by 5 or 10 times for clarity purposes.”

Fig 3, caption: this is the FTIR spectrum of the residue of SO₂/CH₄, not the ice itself.

Reply: Clarified that it is the FTIR of the ice experiments. This now reads:

“Fig. 3. FTIR spectra of sulfur dioxide (SO₂)/methane (CH₄) ice experiments at 320 K after irradiation for 1 hour.”

P5, line 114-119: While the second sentence claims the formation of functional groups of ASAs, the first sentence seems to suggest that ASAs are identified. I would suggest that the authors in sentence one already state that functional groups linked to ASAs are identified.

Reply: To clarify this we have added associated to the first sentence and changed “linked” to “related to” as we feel it emphasizes ASAs are not discretely identified.

“Most importantly, several peaks can be linked to functional groups associated with ASAs such as OH stretching modes at 2899 cm^{-1} and $2523\text{--}2588 \text{ cm}^{-1}$, the CH bending mode at 1464 cm^{-1} , the S=O stretching mode at 1210 cm^{-1} , the SOH bending at 1119 cm^{-1} , CH rocking at 996 cm^{-1} , and the C–S stretching at 724 cm^{-1} (medium dose) and 786 cm^{-1} (high dose)⁴²⁻⁴⁵. Therefore, these data provide compelling evidence that *functional groups related to ASAs* originate during the radiation exposure at 10 K.”

P5, line 121-123: similar to the above point, this sentence suggests that ASAs are detected by their IR signatures, while only IR features indicative of functional groups linked to ASAs are present. I'd suggest that the authors look at the phrasing of this and other sentences to clarify that IR features of functional groups that can be linked to ASAs are present, but not that ASAs are directly detected.

Reply: We have added clarification that these are only associated with ASAs.

“Here, evidence for functional groups associated with ASAs is supported by the OH stretching at 2845 cm^{-1} and 2426 cm^{-1} , the CH bending modes at 1416 cm^{-1} (medium dose) and 1440 cm^{-1} (high dose), the asymmetric S=O stretching at 1297 cm^{-1} , the SOH bending mode at 1186 cm^{-1} , the symmetric S=O stretching at 1116 cm^{-1} , SO stretching at 864 cm^{-1} , and the C-S stretching mode at 771 cm^{-1} .⁴²⁻⁴⁵”

P6, line 136: The authors claim that 22+/-9% of the functional groups remain after warm-up from 10 to 320 K, which I read as: the remaining ~78% has sublimated. IR absorption bands often change (that is, can decrease) due to heating and rearrangement of the molecular structures, which can therefore also account for the decrease in absorbance. I would suggest omitting the statement about the remaining % to avoid confusion about this point and simply state that IR signatures of ASAs functional groups are still present at 320K.

Reply: We have removed the statement regarding the remaining percent of functional group intensity.

P6, line 144: A comparative statement is made about the experiments with and without H₂O. I do not understand why the Hagen et al., 1981 paper is cited here and suggest to remove it.

Reply: Hagen et al was removed from this sentence and moved to the sentence prior (as a source for the fundamental vibrational modes of water).

Fig S6: In the caption, the green fit line for reagents is described, but the green fit is not used in the figure. I would therefore remove it from this caption. Please also check other figures for similar instances.

Reply: This has been removed from Figure 3 and Figure S6.

P9, line 221-222: “these radicals can undergo addition to the sulfur atom via barrierless addition with a barrier of 1 kJ mol^{-1} (Fig. S8)”. Barrierless addition with a barrier seems like a contradiction and requires further explanation. Furthermore, I cannot identify the 1 kJ mol^{-1} barrier in Fig. S8.

Reply: We have clarified this reaction is barrierless for the methyl radical but barriered for the hydroxyl radical. We have also clarified Figure S8 is specifically for the methyl radical addition to SO₂. See below:

“If the methyl and hydroxyl radicals are in proximity to the sulfur dioxide molecule, these radicals can undergo barrierless addition⁵¹ (Figure S8) or barriered addition (1 kJ mol^{-1})⁵², respectively.”

P14, line 375-378: Can the authors provide more details on how the dose is calculated? Specifically, are these doses determined for pure or mixed ices, and was this for the ~1um thick

ice film? If these are calculations for pure ice, how does the dose change in mixed ice?

Reply: This section has been updated to include more information about the mixed ice density and the corresponding physical parameters utilized to calculate this dosage. We have added the effective dosage in both SO₂/CH₄ ices and SO₂/CH₄/H₂O ice experiments to reflect this change.

“According to CASINO simulations, the high dose experiment corresponds to doses of 170 ± 30 eV molecule⁻¹ and 57 ± 9 eV molecule⁻¹ for SO₂ and CH₄, respectively, in the SO₂/CH₄ experiments and 210 ± 30 eV molecule⁻¹, 70 ± 11 eV molecule⁻¹, and 66 ± 11 eV molecule⁻¹ for SO₂, CH₄, and H₂O, respectively, in the SO₂/CH₄/H₂O ices. These doses scale linearly with current and the calculations use the pure ice with a thickness of 1 μm, and incorporate the corresponding mixed ice density. The density of each ice component (SO₂, 1.395 g cm⁻³ ⁷³; CH₄, 0.47 g cm⁻³ ⁶⁷; H₂O, 0.97 g cm⁻³ ⁷⁴) in their respective ratios in the ice were utilized to calculate the final mixed ice density (SO₂:CH₄, 2:5 ± 2, 0.73 ± 0.05 g cm⁻³; SO₂/CH₄/H₂O, 1:2:10 ± 1, 0.90 ± 0.01 g cm⁻³).”

Reviewer #2 (Remarks to the Author):

The study by Mason McAnally et al. explores the abiotic formation of alkylsulfonic acids in interstellar sulfur-doped ices, which are believed to have played a crucial role in early Earth biochemistry. Laboratory simulations demonstrated that these acids can be efficiently produced through the interaction of sulfur dioxide, water, and methane ices with galactic cosmic rays, mimicking conditions in cold molecular clouds. This finding has significant astrophysical implications, suggesting that alkylsulfonic acids could have been synthesized in interstellar environments and delivered to early Earth via comets and asteroids, providing a soluble source of organosulfur compounds essential for the development of prebiotic chemistry. The identification of these acids in meteorites and the carbonaceous asteroid Ryugu supports this hypothesis, highlighting their potential role in the origins of life and the sulfur chemistry in extraterrestrial environments.

The article is very well written and the experimental results is well presented and discussed. I recommend publication after the authors address the following comments:

My main comments are:

1) A discussion about eventual contamination during ex-situ GC-MS analysis and sample holding is needed.

How were the samples stored at room temperature and for how long?

Reply: The identification of alkylsulfonic acids in room-temperature residues of the irradiated ices was compared against a procedural blank to confirm that the acids were formed during ice irradiation and to exclude contamination during storing or ex-situ analysis. The samples were stored until all residues were produced under dry conditions at stable temperature of 22 °C.

We have clarified their storage in the GC×GC-TOF-MS methods section.

“Solid residues that remained after the TPD phase were stored under dry conditions and then extracted *ex situ* with 10 × 50 µL of Milli-Q water (Milli-Q Direct 8, 18.2 MΩ cm at 298 K, < 2 ppb total organic carbon) and transferred into conical reaction vials (1 mL, V-Vial Wheaton).”

2) A discussion about the timescale on astrophysical environments concerning the employed electron beam is needed.

3) Authors should also clarify the readers about the link between the employed electron beam and cosmic rays (with a discussion also about the different penetration depth of both projectiles within matter, and eventual different physicochemical products)

Reply: We have a statement about the link between energetic electrons from the electron beam and their connection to galactic cosmic ray-initiated electron cascades and the timeline these experiments reflect.

“These conditions replicate the processing of interstellar ices in cold molecular clouds by galactic cosmic ray-initiated electron cascades over the lifetime of cold molecular clouds up to 5×10^7 years¹⁸.”

We have opted to include a discussion about the physicochemical effects of GCRs and their generated electron cascades by adding:

“Galactic cosmic ray MeV particles penetrate through icy mantles and the grain core, depositing portions of their energy into interstellar grains. Simulations suggest the energy predominantly results in ionization, releasing energetic electrons born of a few keV^{36,37}. These secondary electrons can penetrate the ice, initiating non-equilibrium chemical reactions. Although these electrons may not completely penetrate through the ice, they can process significantly more material. Here, the electron beam replicates these electron cascades found in various ages of molecular clouds: a low dose simulating 10^6 years in cold molecular clouds (up to 4.2 eV molecule⁻¹), a medium dose (10^7 years; up to 42 eV molecule⁻¹), and a high dose (5×10^7 years, up to 210 eV molecule⁻¹).”

4) Several years ago, Professor Pilling's group conducted studies using X-rays on ices containing SO₂ (Pilling & Bergantini, 2015, *The Astrophysical Journal*, 811(2), 151; Bonfim et al., 2017, *Phys. Chem. Chem. Phys.*, 19, 26906-26917) and the implantation of S ions in ices (Boduch et al., 2015, *Journal of Physics: Conference Series*, 629, 012008; Lv et al., 2014, *Mon. Not. R. Astron. Soc.*, 438, 922-929). These studies could provide valuable insights into the processing of SO₂-rich ices due to ionizing radiation.

The authors may consider citing these studies in their current publication to clarify this aspect for the reader.

Reply: We added previous work in the introduction to make note of these publications and others.

“Additionally, previous experiments have probed SO₂-containing ices²⁷⁻³⁰ as well as sulfur atom implantation³¹⁻³³ to understand the underlying chemistry of sulfur in astrophysically relevant environments.”

5) Authors should add some lines about the possibility of others reaction routes in the formation of the important Alkylsulfonic acids and its precursors.

Reply: We have presented alternate reactions outside of the system resulting in the formation of ASAs.

“Alternatively, reactions leading to ASAs and their precursors could form from other chemical species. Material such as hydrogen sulfide presents a promising starting point, whereby successive oxidation by water or carbon dioxide (CO₂) in methane-containing ices could produce

ASAs.”

6) What are the red lines in figure 5? Models?

Reply: We have clarified that these are from the kinetic fits.

Reviewer #3 (Remarks to the Author):

This paper is generally well written and the results are well presented. A pdf is provided with suggestions for improvement.

Line 36 “How do we know early sulfate deposits have not been reduced by subsequent biology or other geochemical mechanisms?”

Reply: There is some evidence of biological sulfate reduction which does not reflect reduction by geochemical mechanisms (Shen et al., Earth and Planetary Science Letters, 2009). Although, this process of sulfate reduction may occur during the early Archean period, the limited availability of sulfates would constrain the biological alteration of sulfates.

Line 37 “Source of what?”

Reply: We have clarified this is a “source of sulfur”.

Line 41. Bile seems like a late invention for this discussion.

Reply: These chemicals were specifically mentioned for their connection to contemporary biology and their critical function in biology.

Line 56. “Which of these is an enzyme?”

Reply: 3'-phosphoadenosine-5'-phosphosulfate is a coenzyme and as such has been corrected in the text.

Line 323. “Could this be insignificant in terms of mass?”

Reply: Any evidence that sulfur could be trapped on grain surfaces is evidence that it could contribute to depletion of gas-phase sulfur in dense and diffuse molecular clouds. This phenomena is poorly understood and our results indicate organo-sulfur molecules may contribute at least partially to this depletion.

Line 336. “And presumably others?”

Reply: This is correct. We have clarified this could occur in other systems outside of our own.

“Considering that molecular clouds are nurseries of stars and planetary systems, the detection of alkylsulfonic acids proposes that these molecules could have been at least partially delivered to our Solar System and others from their interstellar nurseries via circumstellar disks.”

Line 391. “The solutions were dried under a gentle stream of dry nitrogen.” What made it gentle?

Reply: A Six Port Mini-Vap Evaporator/Concentrator (Merck) was used to control the flow of nitrogen during sample drying. A digital flow meter (Vögtlin Instruments) was used to monitor the flow rate in addition to visually checking that the liquid level in the reaction vial was not

significantly disturbed by the nitrogen flow. Approximately 500 μl of MilliQ® with the extracted residues were dried in about 1.5 hrs.

Line 393. “How was “fully-dried” determined?”

Reply: Upon extraction, the samples were transferred to 1 ml conical bottom reaction vials (V-Vials, Wheaton), which were used for all further manipulation. The conical shape enables to visually monitor the process of sample drying accurately, i.e. the flow of nitrogen was stopped as soon as the last drop at the bottom of the cone evaporated and there was visually no liquid left.

In the spirit of clarity, we have opted to simplify this to “dried” and now reads:

“Subsequently, 20 μL of 10^{-6} M methyl laurate (internal standard, Sigma Aldrich, purity \geq 97.0%) in hexane (Sigma Aldrich, \geq 99.0%) was added to the reaction vials and the mixtures were dried.”

REVIEWERS' COMMENTS

Reviewer #1 (Remarks to the Author):

I thank the authors for addressing my questions. I am happy to recommend their manuscript for publication.

Reviewer #2 (Remarks to the Author):

The authors have addressed all of my comments and improved the manuscript accordingly. I have no further comments and recommend the acceptance of this publication.